# Spatial modeling and ecological suitability of Ebola virus disease in Africa

**Lombo Baluma Didier**[1,2]*, **Lukusa Lumu Jude**[3], **Esuka Igabuchia Franck**[4],
**HaoNing Wang**[5,6], **Xiao-Long Wang**[1,2]

1 Key Laboratory of Wildlife Diseases and Biosecurity Management of Heilongjiang Province, Harbin, Heilongjiang Province, P. R. China, 2 College of Wildlife and Protected Area, Northeast Forestry University, Harbin, Heilongjiang Province, P. R. China, 3 Centre for Research in the Humanities, Ministry of Scientific Research and Technology, Kinshasa, Democratic Republic of the Congo, 4 Health and Environment Option, Department of Public Health, Faculty of Medicine and Pharmacy University of Kisangani, Kisangani, Democratic Republic of the Congo, 5 School of Geography and Tourism, Harbin University, Harbin, Heilongjiang Province, P. R. China, 6 Heilongjiang Cold Region Wetland Ecology and Environment Research Key Laboratory, Harbin University, Harbin, Heilongjiang Province, P. R. China

* lombobaluma@gmail.com

**Data Availability Statement:** The data underlying the results presented in the study are available from https://www.worldclim.org/data/worldclim21.html and https://hub.worldpop.org/project/categories?id=18.

## Abstract

This paper looks into the MaxEnt model in a trial to comprehend the ecological and environmental conditions that propagate and drive the spread of Ebola Virus Disease in Africa. We use the MaxEnt model to assess risk determinants associated with the occurrence and distribution of EVD, taking into account non-correlated variables such as neighborhood mean temperature, rainfall, and human population density. Our findings indicate that among the factors that significantly shape the geographical distribution of EVD risk are human population density, annual rainfall, temperature variability, and seasonality. The model used is both reliable and accurate (the average value for training AUC was 0.987); it can be used as a valuable approach for the prediction of infectious disease outbreaks. High-risk areas are primarily identified in the western and central regions of Africa, with some of the others in the east also vulnerable. This further calls for specified public health interventions and enhanced surveillance in specified hotspots, contributing to global efforts to predict and mitigate risks associated with EVD outbreaks more adequately. The findings further support that it remains imperative to conduct additional research, including socio-economic and cultural variables, to enhance the understanding of how environmental factors contribute to the emergence and transmission of Ebola.

## 1. Introduction

Ebola, also known as Ebola Virus Disease (EVD) and Ebola Haemorrhagic Fever (EHF), is a serious disease caused by the Ebola virus, which belongs to the filovirus family. The disease was first identified in 1976 when two epidemics occurred simultaneously. One in Nzara, a town in southern Sudan, and the other in Yambuku, a village near the Ebola River, after which the disease is named [1–4]. EVD is a deadly viral hemorrhagic fever that occasionally breaks

**Funding:** The author(s) received no specific funding for this work.

**Competing interests:** The authors have declared that no competing interests exist.

out, mostly on the African continent. It is a disease of humans and non-human primates [4, 5]. These include monkeys, gorillas and chimpanzees. The first symptoms are usually fever, sore throat, muscle pain, and headaches [6, 7]. Ebola virus infection in human typically presents with symptoms such as vomiting, diarrhea, rash, and decreased liver and kidney function. As the disease progresses, some individuals may experience internal and external bleeding [5–7]. There are six species of Ebola virus, four of which are known to be the cause of disease in humans: Zaire ebolavirus, Sudan ebolavirus, Taï forest ebolavirus and Bundibugyo ebolavirus. The remaining two species, Reston ebolavirus and Bombali ebolavirus, have not been reported to cause disease in humans [1, 2, 8]. In 2018, Bombali virus was first identified in bats in Sierra Leone. It is currently unknown whether the virus causes disease in animals or humans [9].

The virus is transmitted by direct contact with bodily fluids such as infected human or animal blood or objects recently contaminated with infected fluids [7, 10]. There is no documented evidence for airborne spread between humans or other primates, either in nature or under laboratory conditions [8]. Sexual contact with someone who is sick with or has recovered from EVD can also transmit the virus. People can continue to carry the virus in their semen or breast milk for weeks or months after recovering from Ebola [11, 12]. Flying foxes are thought to be natural carriers of the virus and can transmit the virus while remaining unaffected. Symptoms of Ebola may be similar to those of other diseases such as malaria, cholera, typhoid, meningitis and other viral hemorrhagic fevers [7].

Diagnosis is confirmed by testing blood samples for the presence of viral RNA, viral antibodies, or the virus itself [8, 13, 14]. The disease is often fatal, with a fatality rate ranging from 25% to 90%, about 50% on average depending on the virus species and other factors. Death is often due to shock from fluid loss, and typically occurs between 6 and 16 days after the first symptoms appear [7, 8].

Recent innovations in the applications of SDM have illuminated the dispersal of several zoonotic infections, and extrapolation to Ebola is feasible. These modeling techniques have been used for studies that are somewhat analogous, particularly regarding Lyme disease and Hantavirus, concerning the vector-host-environment scenario [15, 16]. The models, such as MaxEnt, define potential risk areas and are especially useful when planning targeted public health initiatives. To compare the predictive power of the MaxEnt model with other epidemiologic models and validate their effectiveness in forecasting disease.

Several such studies related to the present one are: Mapping the zoonotic niche of Ebola virus disease in Africa [17], Assessing the zoonotic risk of Ebola virus disease in West Africa using a spatial modeling framework [18], and Mapping of Ebola virus spillover: Suitability and seasonal variability at the landscape scale [19]. All these three papers used multicriteria decision analysis (MCDA) methods integrated in GIS; the last two papers are limited to only a few African countries. This paper, however uses a different methodology and has constructed a model which is extendable by including additional variables.

This study is essential as the spatial modeling techniques have been applied to predict EVD outbreaks, which has shown potential in another zoonotic disease. Epidemiological studies applying Species Distribution Models, such as MaxEnt, provide a robust framework for identifying the drivers of disease spread from environmental and socio-demographic factors [20, 21].

Thus, the research described here extends the prior work of these and other authors, who had either worked on more focused geographical scales or applied less well-validated modeling methods. This present study, therefore, uses Maxent to develop an EVD risk model in Africa under different environmental variables influencing the transmission of this disease. This work intends to (1) map distribution patterns of known EVD, (2) make predictions on possible at-risk areas, and (3) identify dominant environmental variables associated with EVD

outbreaks so both national and international policymakers can target appropriate prevention efforts and implement effective surveillance strategies.

## 2. Materials and methods

### 2.1. Study area

Acknowledge Africa as the second-largest and second-most-populous continent in the world. Bring out its rich diversity, represented by more than 1.3 billion people speaking over 2,000 languages [22, 23]. This continent consists of 54 sovereign countries and several territories and regions. Finally, Africa hosts vast landscapes, which include deserts such as the Sahara, tropical rainforests like the Congo, and massive rivers, for example, the Nile, Congo, and Niger rivers [22]. The northernmost point is about 37° N latitude, Ras ben Sakka in Tunisia. The southern end is about 34° 50' S, Cape Agulhas in South Africa. On the other hand, the continent's easternmost point is 51° 27' E, Ras Hafun in Somalia, while the westernmost point is 17° 33' W, Cap Vert Peninsula in Senegal. The Ebola virus disease continues spreading in parts of Africa, provoking panic and affecting morbidity globally. Information on the latest developments in outbreak areas is updated continuously by WHO [11].

### 2.2. Occurrence points

Occurrence data for EVD were sourced from the records of WHO and CDC, reporting cases in Africa between 1976 and 2022–128 records have been published [7, 11]. A minimum distance filter of 10 km between each pair of occurrence points was used to reduce spatial autocorrelation, implemented with the SDM Toolbox integrated into ArcGIS 10.5 [24, 25]. This significantly minimizes spatial sampling bias, making the dataset more robust for modeling.

**2.2.1. Spatial autocorrelation and minimum distance selection.** Spatial autocorrelation can result in overestimating predictive ability if records are closer together than the data's inherent resolution and the residuals between records are correlated [24, 25]. We corrected spatial autocorrelation using the SDM Toolbox v1.1c within ArcGIS 10.5 to increase the robustness of our models. To achieve this, we left a separation distance of 10 km between occurrence points. This distance was selected because any two points were surely representative of two different ecological niches, precluding the possibility of repeating similar environmental conditions in one place [25]. This is critical in heterogeneous areas of the landmass, where the circumstances of environment and ecology sometimes show a sharp variance over small distances [24].

This also spaces the occurrence data points at a 10 km distance, allowing for a more spatial spread and avoiding overestimation in areas with high data accumulation. This strategic spacing of data points leads to a robust model that does not overfit, similar to methodologies in other studies on zoonotic diseases and ecological modeling that supports distances of the exact nature to reduce spatial autocorrelation [24, 25]. Furthermore, a 10 km filter distance is the proper compromise between computational requirements and model accuracy. Lower distances may simply make more calculations necessary and, on the other hand, tend to over-fit models; higher distances can decrease a model's sensitivity to small environmental changes [24, 25].

**2.2.2. Impact on model stability and accuracy.** The application of minimum distance is taking a new shape with multiple effects that it will have on the model. Points divided by a distance of 10 km help to diminish spatial bias within the model, thereby giving more credible predictions. This spacing enables a better representation of different environmental conditions [24].

The reduction in spatial autocorrelation improves the stability of the model in its predictions. Models built with spatially independent data are, to a lesser extent, prone to overfitting and more likely to generalize well with new data [24, 25]. Although this minimum distance setting is characterized by increased model accuracy, concerns have been raised about its ability to make sure that the occurrence points used during calibration genuinely represent actual environmental gradients that drive species distribution. It will help in generating an accurate risk map for EVD outbreaks.

## 2.3. Environmental variables

The two primary environmental datasets were the WorldClim database for climate variables (version 1.4, years 1950–2000) and the WorldPop dataset for human population density (years 2000–2020). These datasets had extensive coverage and high resolution—30 arc-seconds grid cells [25].

Multicollinearity and independence among the variables were tested using the Statistical Software for the Social Sciences (SSPS v22.0) with 68 factors; after this step, 22 factors were retained. The stepwise selection was done to arrive at a robust and reliable model. Dimensional reduction was done by carrying out a principal component analysis. Only those variables with eigenvalues over 0.97 and forming visual inflection points in the scree plot were retained [26, 27]. Besides, excluding those variables with VIF over ten was done to minimize multicollinearity in the model [25, 27]. This strict selection has limited us to six core variables—those that mean the most independence for relevant predictive power for model: Mean Temperature of the Warmest Quarter, Annual Mean Temperature, Max Temperature of the Warmest Month, Mean Temperature of the Driest Quarter, Precipitation of the Driest Quarter, Precipitation of the Driest Month. The MaxEnt algorithm was further processed for the remaining bioclimatic variables.

The relative contribution of the predictors in the model was then understood through three different analytical methods: jackknife contribution, response curve analysis, and backward stepwise elimination method. Variables contributing less than 3% were systematically dropped from the model, which allowed the inclusion of only the most significant predictors for analysis [28]. Fig 1 below presents a scree plot from our principal component analysis, which illustrates the eigenvalues of the principal components and helps determine the number of components to retain for further analysis.

The following scree plot shows the eigenvalues of principal components for a PCA. A pronounced, rapid decay beyond the first component combined with leveling off past the third component usually indicates that the first few components capture most of the variance in the data. This in turn, suggests that it will probably be sufficient to retain the first three or four components to obtain a helpful summary of the dataset. Further components each contribute very little to the explanation of further variance.

The first six principal components of Table 1 account for a total of 86.91% variance of the original data set. This significant dimensionality reduction would, therefore, help reduce complex analysis while maintaining most of the critical information innately present within the data. We capture the primary structure of the dataset by focusing attention on these components, making the analytical process much more streamlined yet as compelling as ever.

Table 2 presents the variance inflation factors (VIF) for the bioclimatic variables used in the model. Notably, several temperature-related variables exceeded the VIF threshold of 10, indicating significant multicollinearity. For these variables, the VIFs computed are way above the threshold often used at ten and, therefore can be said to be not independent from one another. This is why we did not consider all variables with a VIF value above 10.

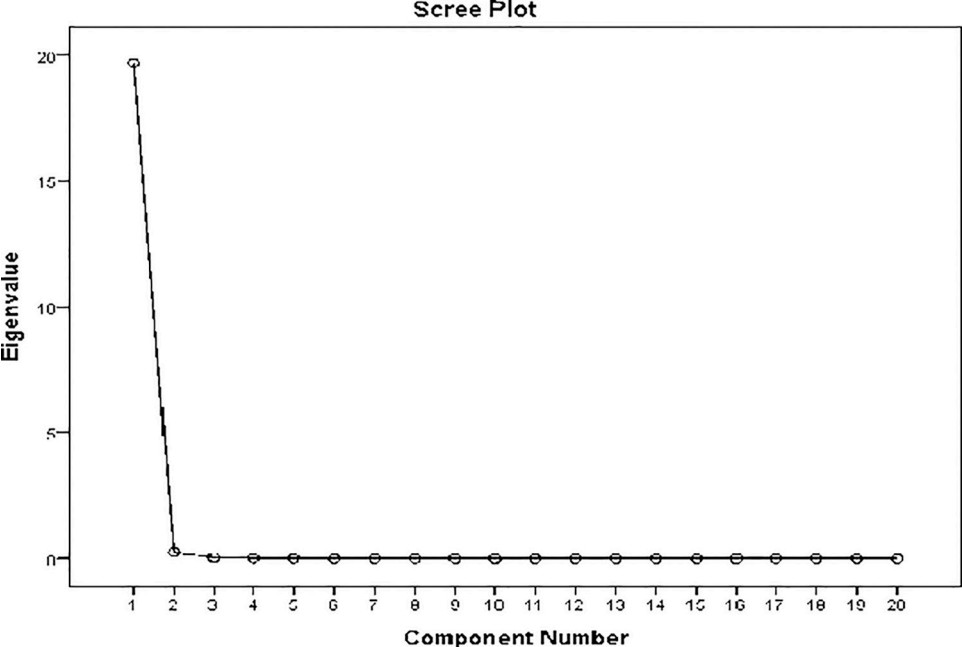

**Fig 1. Scree plot of principal component analysis.**

In Table 3, 'Population density' and 'Temperature seasonality' emerge as some of the variables with very high general impacts, either through direct contribution or permutation importance over the model. 'Population density' seems to be the most important predictor, while 'Temperature seasonality' significantly affects model accuracy. Other variables include 'Precipitation of the driest quarter' and 'Annual precipitation.'

## 2.4. Model building

We adopted the Maximum Entropy (MaxEnt) modeling approach because it is one of the best ways to implement ecological niche models, even when dealing with small amounts of data [26, 28]. The MaxEnt model run was allowed a maximum of 5000 iterations during its run, with the convergence threshold set at 0.00001, and up to 5000 background points were run. A 70/30 splitting approach was used when partitioning between training and test datasets for testing the predictive performance of the model. Presence pseudo-absence locations were randomly located throughout the model study area to represent potential background points, enhancing model discrimination capability between suitable and unsuitable habitats for EVD [26, 28].

We also used the jackknife test to determine which variables are least/most important, hence for reduced area tests. We used response curve analyses to determine the sensitivity of

**Table 1. Principal component analysis summary of explained variance ratios.**

| Principal Component | Explained Variance Ratio | Cumulative Explained Variance |
|---|---|---|
| PC1 | 0.3161 | 0.3161 |
| PC2 | 0.2098 | 0.5259 |
| PC3 | 0.1363 | 0.6622 |
| PC4 | 0.0935 | 0.7557 |
| PC5 | 0.0671 | 0.8228 |
| PC6 | 0.0463 | 0.8691 |

**Table 2. Variance inflation factors for bioclimatic variables in ecological modeling.**

| Variable | VIF |
| --- | --- |
| Annual Mean Temperature | 2071.46 |
| Mean Temperature of Warmest Quarter | 2454.10 |
| Mean Temperature of Coldest Quarte | 1801.64 |
| Temperature of Coldest Month | 50.16 |
| Precipitation of Wettest Month | 112.39 |

predictions to the threshold chosen [30]. We identified the suitability and unsuitability threshold using the criterion that gives the most significant sum of sensitivity and specificity. This threshold optimization is essential to ensure that the model's prediction closely resembles the observed distribution patterns of EVD, hence reliable maps for public health planning and resource allocations [30, 31].

## 2.5. Model evaluation

Traditionally, the Area Under the Receiver Operating Characteristic Curve AUC has been the primary metric for assessing the discriminatory capability of ecological niche models, indicating how well the model can differentiate between presence and absence across all probability thresholds. For our model, the AUC was notably high, averaging 0.987, reflecting exceptional discrimination, thus, it performed very well [29]. Although robust, the AUC is sensitive to the choice of threshold for classifying suitable and unsuitable conditions.

The True Skill Statistic was reported along with AUC for the model. It balances sensitivity against specificity but does not depend on prevalence. Our model's TSS was 0.8039, which is very good in the perspective of any ability to avoid omission as well as commission errors—important conditions for effective risk mapping in public health [30].

We also used the F1 score, which is the harmonic mean of precision and recall. This measure is of especial use when an uneven distribution in a class could make other measures not as informative. Our model reached an F1 Score of 0.8913, underpinning accuracy to predict true-presence locations of Ebola Virus Disease [31].

These metrics collectively indicate that our model is not only capable of high accuracy but also balanced in terms of error handling, making it a reliable tool for forecasting disease risk and aiding in the strategic planning of health interventions.

## 3. Result

### 3.1. Occurrence points

Analysis of 128 occurrence records from WHO and CDC revealed significant clustering of Ebola cases primarily in the western and central regions of Africa as it shows in Fig 2 below.

**Table 3. Contribution and permutation importance of environmental variables in the maxent model.**

| Variables | Percent contribution | Permutation importance |
| --- | --- | --- |
| Population density | 54.3 | 21.1 |
| Precipitation of driest quarter | 23.6 | 19.3 |
| Annual precipitation | 8 | 2.1 |
| Precipitation of coldest quarter | 4.5 | 7.7 |
| Mean diurnal range | 4.1 | 1.3 |
| Temperature seasonality | 3.4 | 47 |

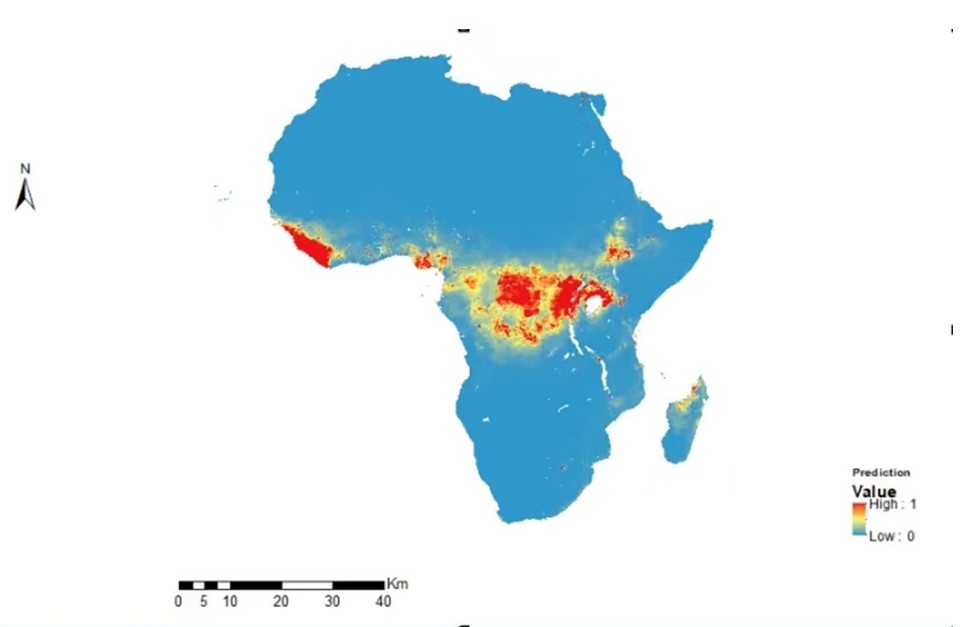

**Fig 2. Ebola virus disease high-risk areas in Africa.**

These clusters correspond closely with areas historically known for frequent outbreaks, providing a robust foundation for the spatial modeling efforts.

We reclassified the MaxEnt spatial model output into three risk classes, namely high, moderate and low risk areas. The map illustrates that the areas with the highest risk for the emergence of Ebola virus are located in the western and central of Africa regions, as well as specific areas in the east. In contrast, some of central Africa is considered to be at a medium risk level, while certain areas of southern and north Africa and Madagascar are classified as low-risk regions.

## 3.2. Environmental variables

These curves show how each environmental variable affects the Maxent prediction (Fig 3). Response curves describe how the predicted probability of presence changes with an increase in an environmental variable while all other environmental variables are kept at their average sample value. The following figure shows the response curve of 6 keys variables that are included in our model.

The response curves for the predictors show that human population density, annual rainfall, rainfall of the coldest quarter, rainfall of the driest quarter, average diurnal range, and temperature seasonality affect the Ebola virus in Africa.

## 3.3. Model Building

The MaxEnt model was run with 5000 iterations at a convergence threshold of 0.00001 and with up to 5000 background points, hence ensuring thorough sampling. This configuration allowed for robust discrimination of suitable and unsuitable habitats, as indicated by preliminary suitability maps created during the model runs.

From the Jackknife test in Fig 4, it is evident that exclusion of any of the six variables yields variations of the model's regularization gain, test gain, and AUC. On single evaluation as a

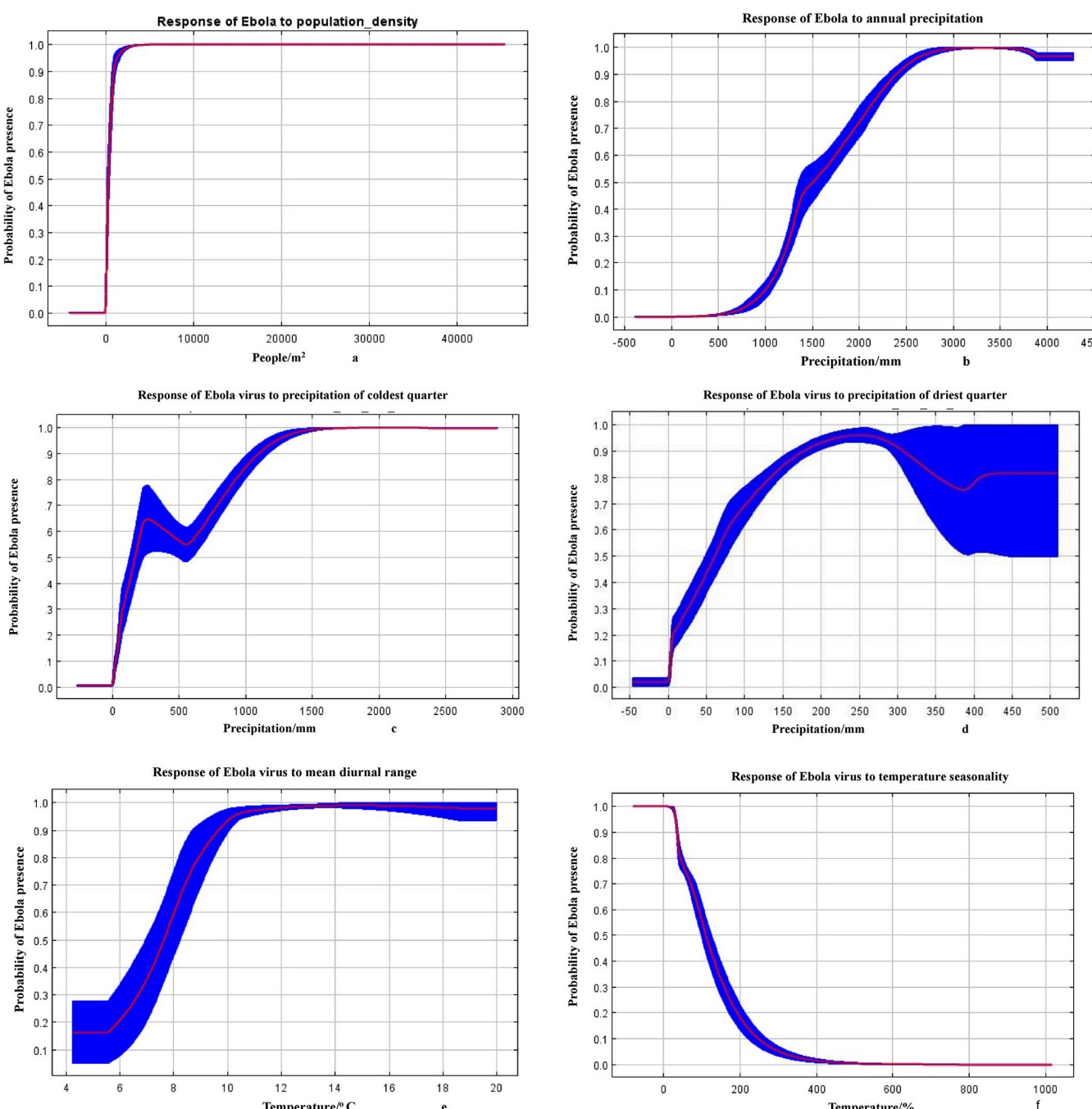

**Fig 3.** Response curse of population density (a) Annual precipitation, (b) Precipitation of coldest quarter, (c), (d) Precipitation of driest quarter, Mean diurnal range (e) and (f) Temperature seasonality for Ebola Virus suitability in Africa. The red lines indicate the mean values, while the blue lines denote the standard deviation.

single environmental variable, population density and annual precipitation had the highest training gains, while the mean diurnal range realized the lowest values. Besides, the minimum training gain was obtained when population density and annual precipitation were excluded, and the maximum training gain was obtained when the mean diurnal range was excluded.

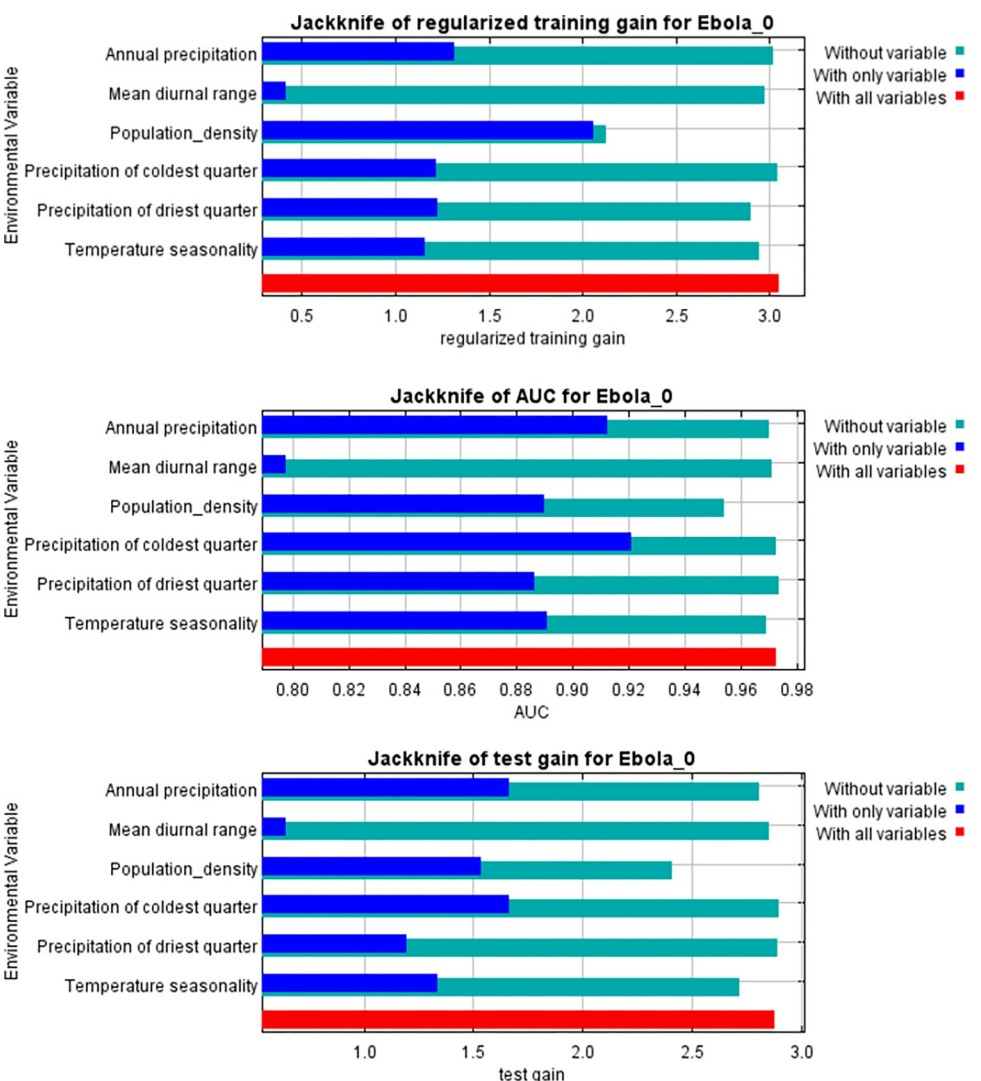

**Fig 4. Summary of jackknife analysis conducted to ascertain the importance of each environmental variable.**

### 3.4. Model evaluation

The model exhibited high predictive accuracy with an average training AUC of 0.987 across multiple runs. The selection of AUC as the primary evaluation metric is justified by its ability to measure the model's discriminatory capacity effectively. Thresholds for suitability were set using a combination of sensitivity and specificity, optimizing the balance for accurate risk mapping."

As shown in Fig 5, the average area under the curve (AUC) for the MaxEnt model across 100 bootstrap replicates was 0.987, demonstrating excellent discriminatory ability. We observed a True Skill Statistic (TSS) of 0.8039. This high TSS value confirms the model's effectiveness in accurately identifying suitable and unsuitable habitats for Ebola, highlighting its practical utility in public health scenarios. Additionally, the F1 Score of 0.8913 further supports the model's precision and robustness in predicting actual presence locations, enhancing its application in designing targeted interventions and resource allocation.

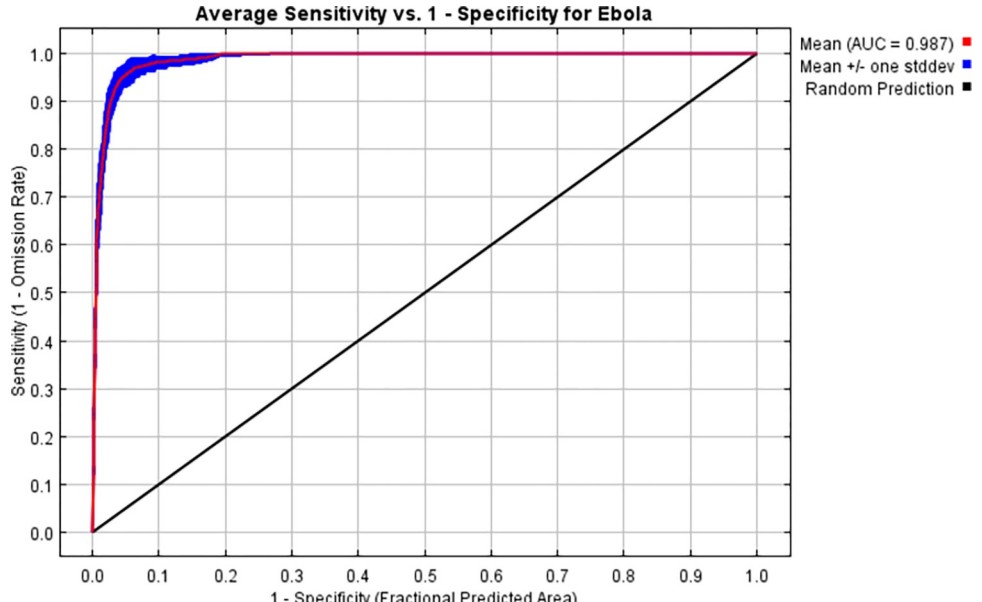

**Fig 5. Average receiver operating characteristics and related area under the curve (AUC) of the 100 bootstrap replicates.**

## 4. Discussion

### 4.1. Limitations of the modeling technique

The applied modeling approach, which integrates geospatial techniques with MaxEnt, presents several limitations. While MaxEnt is a widely used method for species distribution modeling, particularly for rare species with small datasets, it assumes that the species are at equilibrium with the environment, which might not always be true, particularly in highly dynamic ecosystems like those found in our study area. In the model, the environmental and demographic situations are considered fixed. This limitation can be so important because those factors are very dynamic in nature: human migration, changing land use, and climate variability may change the risk landscape for EVD over time [32–35]. The same model applied to another region would need enormous adjustments to take into account local data and, most probably, different environmental and demographic variables. Moreover, the reliance on available environmental variables might introduce biases due to the limited number of climatic factors included in the model. For instance, [36] highlighted the importance of accounting for land-use changes in addition to climatic variables in species distribution models, suggesting that a more comprehensive dataset could enhance prediction accuracy.

Another limitation is the resolution of the geospatial data. The environmental layers used in MaxEnt were sourced from publicly available datasets, which often offer coarse resolutions. This might limit the precision of habitat suitability predictions in microhabitats where fine-scale environmental variation plays a significant role. Furthermore, as identified in studies such as [37], model outputs are sensitive to the choice of regularization parameters, and incorrect parameter settings can either overfit or underfit the data. Future studies could refine these limitations by incorporating high-resolution environmental layers and more robust parameter selection processes, as noted by [36].

## 4.2. Benefits of the applied modeling techniques

Despite these limitations, combining geospatial analysis with MaxEnt provided significant insights into the spatial distribution of the target species. MaxEnt's ability to handle small sample sizes is particularly advantageous in regions like ours, where data scarcity is a significant challenge. The integration of geospatial techniques further allowed for a spatially explicit analysis that identified priority zones for conservation actions, a critical step in adaptive management strategies. This aligns with the findings of [37], where MaxEnt successfully predicted the distribution of tick vectors under various environmental drivers, offering essential data for establishing early warning systems.

The key strength of this modeling approach lies in its ability to identify the most relevant environmental factors influencing species distribution. For instance, our model revealed that population density, temperature variability and precipitation were critical factors, corroborating findings from studies like [37–39], where similar variables were shown to significantly influence the spatial distribution.

In addition, both studies underscore the importance of predictive modeling for future conservation planning. In [38], they used MaxEnt to predict future oak tree distribution ranges under various climate scenarios, providing valuable information for proactive conservation efforts. In a parallel application, [37] employed similar techniques to predict the spatial distribution of *Hyalomma* ticks, demonstrating the utility of MaxEnt in addressing both plant and animal species distributions across different ecosystems. Together, these studies affirm that MaxEnt, when combined with geospatial techniques, is a powerful tool for both current and future ecological planning.

## 4.3. Occurrence points

The significant clustering of cases of Ebola was demonstrated in the analysis of 128 occurrence records from the WHO and CDC in most parts of Africa, mainly west and central. It is essential to underline that these clusters are critically aligned to areas that are historically known for a higher frequency of outbreaks, which supports the efforts of spatial modeling that were done. The MaxEnt model was well informed by the occurrence data and identified hot spots in western and central Africa, with some areas in the eastern part, through empirical evidence [40–42]. This indicates that the model has been confirmed of its validity to intensify the need for focused public health interventions in these regions.

In keeping with historical patterns of Ebola emergence and areas where the virus is endemic, risk levels are geographically concentrated in the western and parts of the eastern regions of Africa. From a public health perspective, this geographic dimension is essential for targeting and surveillance. Research that maps zoonotic niches of Ebola in Africa presents risk profiles that are pretty similar and points to the need for targeted surveillance in risk areas [42].

## 4.4. Environmental variables

The developed response curves describe the importance of some key environmental variables on the likelihood of Ebola presence. The distribution of EVD is highly affected by essential variables, such as human population density, annual precipitation, and temperature seasonality. In addition, the analysis shows high population densities to be a factor correlated with high risk because of likely increased human-to-human transmission opportunities. This finding is in agreement with some previous studies conducted about Ebola's ecology and environment. For example, some studies found that EVD outbreaks are related to human-induced deforestation [43–45]. What is underlined here is the intricate relationship between environmental

changes and disease emergence. Similarly, rainfall and temperature changes have been related to the suitability of habitats for the virus or its vectors [43, 44]. Such studies are essential for deriving focused environmental and health policies for risk reduction.

## 4.5. Model building

Coupled with the systematic selection of predictor variables and extensive iteration up to 5000 times the model is very robust in giving a highly reliable framework to map Ebola risk. In this regard, the choice to use a 70/30 split for training and test datasets ensures not only good training but also validation against unseen data to enhance predictive reliability. When pseudo-absence locations were included, the model could discriminate effectively between suitable and unsuitable habitats, providing a fine-scale understanding of risk distribution across different African regions. This is by studies such as [42]. Further evidence of the applicability of this approach can be obtained from the survey of filoviruses' ecological niche modeling implemented in Uganda [43]. Through a 75/30 split of the training and testing data sets and extensive iteration, the authors of the study validated the ability of their model to differentiate between suitable and unsuitable habitats in a manner that emphasizes the significance of the training phase in an ecological model [42, 43].

The jack-knife test results provide some insight, showing the consequence of excluding some variables on the model's performance. This is indicative of the nonlinearity and complexity in ecological niches for diseases like Ebola. These are combinations of several factors interacting to create such a niche. Similarly, a comparative study identified the critical ecological drivers for Ebola and showed that in addition to climate factors, land use changes and animal reservoirs play an essential role [40, 41]. Several studies have shown essential relations between Ebola outbreaks and tropical rainforest biomes, where potential reservoirs for bats are probably contained [44].

## 4.6. Model evaluation

High predictive accuracy was seen in the MaxEnt model, with an average training AUC of 0.987. The excellent accuracy means that the model discriminates areas at risk from those without risk of EVD across all probability cut-offs. This ability is crucial in public health planning and resource allocation, allowing for preventive action where a high risk may be predicted. Optimization of the threshold for suitability based on sensitivity and specificity ensures that models' predictions resemble, as closely as possible, the actual distribution patterns, making the risk maps reliable and actionable. This is commendable in accuracy and concordant with findings from other studies that have also noted the potential of machine learning and statistical models in infectious disease forecasting [45, 46].

Evaluation of our MaxEnt model to True Skill Statistic and F1 Score gives us in-depth insight into predictive performance and reliability. This model has a TSS of 0.8039, which means the model reasonably balances sensitivity and specificity. Our model compared well with other ecological niche models in epidemiological studies, and this was a further justification of the need to use multiple metrics to assure the model's robustness and applicability across different disease scenarios.

The F1 score of 0.8913 further highlights the precision and robustness of our prediction. It is an important measure that is usually lacking in other studies. In conjunction with the AUC, it would provide a more informed understanding of the ability of a model to properly detect the areas where diseases are truly present. The studies also emphasized the increased relevance of the F1 Score to the infectious disease models, especially those of spatially and temporally varying diseases such as Ebola [47, 48].

Therefore, the use of these metrics in our current study adheres to the suggestion of some authors recommending their use in a multi-metric evaluation approach in order to understand the complexities and subtleties of ecological niche modeling in disease prediction. This will ensure that the model, which performs well with the training data, also generalizes effectively on new data a critical factor for models used in public health application [49].

These metrics together validate our model's suitability for forecasting high-risk areas of Ebola, supporting improved preparedness and effective response strategies. By situating our model within the context of these broader discussions, it becomes clear that the methodological choices we've made particularly in terms of model evaluation place our research at the forefront of current epidemiological modeling practices. This comparative approach not only demonstrates the robustness of our model but also its potential contribution to global health security by enhancing disease surveillance and response capabilities.

## 5. Conclusion

The present study demonstrates that the model framework of MaxEnt helps forecast the risk and distribution of EVD across Africa with a suite of sophisticated environmental and demographic predictors. Using spatial modeling approaches, especially the MaxEnt model, identified human population density, temperature variability, and precipitation as key correlates for EVD occurrence. These results underscore the importance of incorporating spatial modeling in public health strategies to pre-identify high-risk areas and target interventions.

The results of this study support the integration of more extensive socio-economic and cultural factors into future models towards ascertaining a more holistic view of dynamics that affect EVD spread. Such an approach is likely to fine-tune public health responses further and will be critical in developing more effective prevention and control strategies suited to the carefully nuanced ecological and demographic contexts of different African regions.

In addition, further research is warranted to develop the model for use against a broader array of zoonoses, building on the solid framework established here to increase global health security against other emerging infections.

## Author Contributions

**Conceptualization:** Lombo Baluma Didier, Xiao-Long Wang.

**Data curation:** Lombo Baluma Didier, Lukusa Lumu Jude, Esuka Igabuchia Franck.

**Formal analysis:** Lombo Baluma Didier, HaoNing Wang.

**Methodology:** Xiao-Long Wang.

**Resources:** Xiao-Long Wang.

**Software:** Lombo Baluma Didier.

**Supervision:** Xiao-Long Wang.

**Validation:** Lombo Baluma Didier.

**Visualization:** Lombo Baluma Didier, Xiao-Long Wang.

**Writing – original draft:** Lombo Baluma Didier.

**Writing – review & editing:** Lombo Baluma Didier, Lukusa Lumu Jude, Esuka Igabuchia Franck, HaoNing Wang.

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
