## [Decision Letter · Decision Letter 0]

4 Jun 2024

PONE-D-24-12702SPATIAL MODELING AND ECOLOGICAL SUITABILITY OF EBOLA VIRUS DISEASE IN AFRICAPLOS ONE

Dear Dr. LOMBO,

Thank you for submitting your manuscript to PLOS ONE. After careful consideration, we feel that it has merit but does not fully meet PLOS ONE’s publication criteria as it currently stands. Therefore, we invite you to submit a revised version of the manuscript that addresses the points raised during the review process.

We look forward to receiving your revised manuscript.

Kind regards,

Adam R. Aluisio, M.D MSc., DTM&H

Academic Editor

PLOS ONE

Journal Requirements:

2. Please ensure that you refer to Figure 1-3 and 5 in your text as, if accepted, production will need this reference to link the reader to the figure.

3. Please include your tables as part of your main manuscript and remove the individual files. Please note that supplementary tables (should remain/ be uploaded) as separate ""supporting information"" files

4. We note you have included a table to which you do not refer in the text of your manuscript. Please ensure that you refer to Table 1 in your text; if accepted, production will need this reference to link the reader to the Table.

Reviewers' comments:

Reviewer's Responses to Questions

**Comments to the Author**

1. Is the manuscript technically sound, and do the data support the conclusions?

Reviewer #1: Yes

Reviewer #2: Yes

2. Has the statistical analysis been performed appropriately and rigorously? 

Reviewer #1: Yes

Reviewer #2: Yes

3. Have the authors made all data underlying the findings in their manuscript fully available?

Reviewer #1: Yes

Reviewer #2: Yes

4. Is the manuscript presented in an intelligible fashion and written in standard English?

Reviewer #1: Yes

Reviewer #2: Yes

5. Review Comments to the Author

Reviewer #1: The manuscript entitled "SPATIAL MODELING AND ECOLOGICAL SUITABILITY OF EBOLA VIRUS DISEASE IN AFRICA” used the Maxent distribution modeling approach together with relevant environmental variables to map known distribution pattern and predict potential distribution of Ebola virus disease. The manuscript is written well and the drawn conclusions are coherent with the obtained results. Although similar methodologies are common, the results of the study could have useful implications for management actions. The manuscript requires some changes before its ready for publication.

Comments and suggestions:

Introduction

- The introduction is quite wordy, I suggest removing some text, for example, the paragraphs…”The first outbreaks of EVD occurred in isolated villages in central Africa [citation needed]……..”. The next two paragraphs following this are also redundant.

Please try to focus on why this study is important and how it is different from other studies conducted.

- Introduction could benefit from a review on modeling other zoonotic diseases that applied the same techniques as this study.

- Please provide some information on the available modeling approaches or SDMs such as the one this study has used (i.e., Maxent) for epidemiological studies.

- Please outline the objectives of the study clearly at the end of the introduction.

Materials and Methods

2.4. Model development and evaluation

- I suggest restructuring the methodology section into the following subsection:

1- Occurrence points

2- Environmental variables

3- Model building (providing details on model parameterization is also necessary, e.g., information of the background points of the Maxent model)

4- Model evaluation

- “Area Under the Receiver Operating Curve (AUC)”; this metric alone, sometimes, is not sufficient as the only evaluation metric. Please justify why only AUC was used. In this section, it's important to clarify what threshold was used to delineate the suitability and unsuitability areas. These questions should be answered in the methodology section.

Results:

Results are also required to be structured similar to the methods i.e., 1,2,3,4, and each subsection should be reported clearly.

Discussion

Discussion requires two subsection in which the the limitations of the modeling technique should be discussed. In addition to this, a small section highlighting the benefits of the applied modeling techniques, i.e., combing geospatial techniques with Maxent and identifying relevant factor preferences in establishing priority zones for management actions is necessary. For these suggestions I recommend these references:

https://www.mdpi.com/2071-1050/15/18/13669

https://doi.org/10.1016/j.ecoinf.2022.101930

https://doi.org/10.1007/s10661-024-12438-z

Reviewer #2: Ebola prevention is crucial as it effectively reduces outbreaks, protects public health, saves lives, minimizes socioeconomic losses, and promotes global health security. This paper not only contributes significantly to the theoretical understanding of EVD transmission mechanisms but also provides a scientific basis for the formulation of practical public health interventions. Its research methods and results have important application value and relevance in the field of infectious disease prevention and control.

The paper uses the MaxEnt model to evaluate the ecological suitability and spatial distribution of Ebola Virus Disease (EVD) in Africa, an effective and widely used technical approach. However, there are some specific details in the application and description of the technical methods that can be improved to enhance the scientific rigor and reproducibility of the study:

1.Supplement descriptions of data quality control and validation steps.

2.The paper uses SDM Toolbox v1.1c to handle spatial autocorrelation, setting a minimum distance of 10 km. However, there is a lack of detailed explanation regarding the methods and criteria for this spatial autocorrelation treatment. It is suggested to elaborate on the rationale for choosing 10 km as the minimum distance and how this setting affects the stability and accuracy of the results.

3.Provide specific results of PCA and VIF testing.

4.Use more evaluation metrics to comprehensively assess the model's performance.

5.Although the results show significant impacts of population density and rainfall patterns on EVD, the practical application of these results can be further explored. It should be discussed how to translate these findings into practical applications, specifically how public health measures should be formulated and implemented based on these results. For example, how these findings can be specifically applied to public health interventions and policy-making.

6. PLOS authors have the option to publish the peer review history of their article (what does this mean?). If published, this will include your full peer review and any attached files.

Reviewer #1: No

Reviewer #2: No

---

## [Author Response · Author response to Decision Letter 0]

30 Aug 2024

Reviewer #1:

Introduction

1.The introduction is quite wordy, I suggest removing some text, for example, the paragraphs…” The first outbreaks of EVD occurred in isolated villages in central Africa [citation needed] …….”. The next two paragraphs following this are also redundant.

Response: We have enlarged the paragraph starting with the first outbreaks of EVD occurred in isolated villages in central Africa and the two paragraphs that follow this paragraph.

2.Please try to focus on why this study is important and how it is different from other studies conducted.

Response: This study is essential as the spatial modeling techniques have been applied to predict EVD outbreaks, which has shown potential in another zoonotic disease. Epidemiological studies applying Species Distribution Models, such as MaxEnt, provide a robust framework for identifying the drivers of disease spread from environmental and socio-demographic factors [20,21]. 

3.Introduction could benefit from a review on modeling other zoonotic diseases that applied the same techniques as this study.

Response: Several such studies related to the present one are: Mapping the zoonotic niche of Ebola virus disease in Africa [17], Assessing the zoonotic risk of Ebola virus disease in West Africa using a spatial modeling framework [18], and Mapping of Ebola virus spillover: Suitability and seasonal variability at the landscape scale [19]. All these three papers used multicriteria decision analysis (MCDA) methods integrated in GIS; the last two papers are limited to only a few African countries. This paper, however uses a different methodology and has constructed a model which is extendable by including additional variables. 

4.Please provide some information on the available modeling approaches or SDMs such as the one this study has used (i.e., Maxent) for epidemiological studies.

Response: Recent innovations in the applications of SDM have illuminated the dispersal of several zoonotic infections, and extrapolation to Ebola is feasible. These modeling techniques have been used for studies that are somewhat analogous, particularly regarding Lyme disease and Hantavirus, concerning the vector-host-environment scenario [15,16]. The models, such as MaxEnt, define potential risk areas and are especially useful when planning targeted public health initiatives. To compare the predictive power of the MaxEnt model with other epidemiologic models and validate their effectiveness in forecasting disease. 

5.Please outline the objectives of the study clearly at the end of the introduction.

Response: We put the objectives of this study in this end of introduction. This work intends to (1) map distribution patterns of known EVD, (2) make predictions on possible at-risk areas, and (3) identify dominant environmental variables associated with EVD outbreaks so both national and international policymakers can target appropriate prevention efforts and implement effective surveillance strategies.

Materials and Methods

6. I suggest restructuring the methodology section into the following subsection:

(1) Occurrence points

(2) Environmental variables

(3) Model building (providing details on model parameterization is also necessary, e.g., information of the background points of the Maxent model)

(4) Model evaluation

Response: We have restructured the methodological part as requested by reviewer 1 and for point 3 we have added this: We adopted the Maximum Entropy (MaxEnt) modeling approach because it is one of the best ways to implement ecological niche models, even when dealing with small amounts of data [26, 28]. The MaxEnt model run was allowed a maximum of 5000 iterations during its run, with the convergence threshold set at 0.00001, and up to 5000 background points were run. A 70/30 splitting approach was used when partitioning between training and test datasets for testing the predictive performance of the model. Presence pseudo-absence locations were randomly located throughout the model study area to represent potential background points, enhancing model discrimination capability between suitable and unsuitable habitats for EVD [26, 28].

7.“Area Under the Receiver Operating Curve (AUC)”; this metric alone, sometimes, is not sufficient as the only evaluation metric. Please justify why only AUC was used. In this section, it's important to clarify what threshold was used to delineate the suitability and unsuitability areas. These questions should be answered in the methodology section.

Response: The overall performance evaluation was based on the selected primary metric: the Area Under the Receiver Operating Characteristic Curve (AUC). AUC is applicable especially for the measurement of the discriminatory performance of the model, i.e., how well the model can distinguish areas with the presence or absence of EVD for all probability cut-offs. Our model had an average AUC of 0.987; thus, it performed very well [29]. Although robust, the AUC is sensitive to the choice of threshold for classifying suitable and unsuitable conditions. We also used the jackknife test to determine which variables are least/most important, hence for reduced area tests. We used response curve analyses to determine the sensitivity of predictions to the threshold chosen [30]. We identified the suitability and unsuitability threshold using the criterion that gives the most significant sum of sensitivity and specificity. This threshold optimization is essential to ensure that the model's prediction closely resembles the observed distribution patterns of EVD, hence reliable maps for public health planning and resource allocations [30,31].

Results:

8. Results are also required to be structured similar to the methods i.e., 1,2,3,4, and each subsection should be reported clearly.

Response: The result was arranged according to this suggestion and each session was very well clarified.

Discussion

9.Discussion requires two subsections in which the the limitations of the modeling technique should be discussed. In addition to this, a small section highlighting the benefits of the applied modeling techniques, i.e., combing geospatial techniques with Maxent and identifying relevant factor preferences in establishing priority zones for management actions is necessary. For these suggestions I recommend these references:

10.https://www.mdpi.com/2071-1050/15/18/13669

11.https://doi.org/10.1016/j.ecoinf.2022.101930

12.https://doi.org/10.1007/s10661-024-12438-z

Response: We have inserted two sessions into the discussion, the first is 4.1. Benefits of the Applied Maxent Modeling Techniques and the second Limitations of the MaxEnt Modeling Technique. We are then adapting our discusion, taking into account the links to the papers sent to us by the reviwiers.

Reviewer #2:

1.Supplement descriptions of data quality control and validation steps

Response: In order to provide further clarity regarding the quality control of our data and the different steps of validation, we have added an additional description in Line ?-?.

2.The paper uses SDM Toolbox v1.1c to handle spatial autocorrelation, setting a minimum distance of 10 km. However, there is a lack of detailed explanation regarding the methods and criteria for this spatial autocorrelation treatment. It is suggested to elaborate on the rationale for choosing 10 km as the minimum distance and how this setting affects the stability and accuracy of the results.

Response: We've added two sub-sessions to the occurrence point session to explain why we've chosen 10km as the minimum for the use of SDM Toolbox v1.1c to handle spatial autocorrelation. Please refer to Line ?-?.

3.Provide specific results of PCA and VIF testing.

Response: Two additional tables have been incorporated into the third section of the methodology. One table presents the results of the principal component analysis (PCA) (Table 1), while another displays the results of the variance inflation factor (VIF) analysis (Table 2).

4.Use more evaluation metrics to comprehensively assess the model's performance.

Response: Thank you for your valuable suggestions. Based on your feedback, we have included additional evaluation metrics to comprehensively assess the model's performance. Below are the specific improvements we made and the results obtained:

Improvements

5.Although the results show significant impacts of population density and rainfall patterns on EVD, the practical application of these results can be further explored. It should be discussed how to translate these findings into practical applications, specifically how public health measures should be formulated and implemented based on these results. For example, how these findings can be specifically applied to public health interventions and policy-making.

Response: The results were discussed in detail in Line, demonstrating how our findings could be applied to public health by assisting decision-makers in implementing effective and responsible measures to manage the population in order to prevent the outbreak of the Ebola virus in the future.

For your convenience, we paste it below:

---

## [Decision Letter · Decision Letter 1]

24 Sep 2024

PONE-D-24-12702R1SPATIAL MODELING AND ECOLOGICAL SUITABILITY OF EBOLA VIRUS DISEASE IN AFRICAPLOS ONE

Dear Dr. LOMBO,

Thank you for submitting your manuscript to PLOS ONE. After careful consideration, we feel that it has merit but does not fully meet PLOS ONE’s publication criteria as it currently stands. Therefore, we invite you to submit a revised version of the manuscript that addresses the points raised during the review process.

We look forward to receiving your revised manuscript.

Kind regards,

Clement Ameh Yaro, Ph.D

Academic Editor

PLOS ONE

Journal Requirements:

Reviewers' comments:

Reviewer's Responses to Questions

**Comments to the Author**

1. If the authors have adequately addressed your comments raised in a previous round of review and you feel that this manuscript is now acceptable for publication, you may indicate that here to bypass the “Comments to the Author” section, enter your conflict of interest statement in the “Confidential to Editor” section, and submit your "Accept" recommendation.

Reviewer #1: (No Response)

Reviewer #2: All comments have been addressed

2. Is the manuscript technically sound, and do the data support the conclusions?

Reviewer #1: Partly

Reviewer #2: Yes

3. Has the statistical analysis been performed appropriately and rigorously? 

Reviewer #1: N/A

Reviewer #2: Yes

4. Have the authors made all data underlying the findings in their manuscript fully available?

Reviewer #1: Yes

Reviewer #2: Yes

5. Is the manuscript presented in an intelligible fashion and written in standard English?

Reviewer #1: Yes

Reviewer #2: Yes

6. Review Comments to the Author

Reviewer #1: The authors have partly addressed my concerns. The authors still need to address the comments that I provided for the 'Discussion' section with citing the relevant references provided.

" Discussion requires two subsections in which the the limitations of the modeling technique

should be discussed. In addition to this, a small section highlighting the benefits of the applied

modeling techniques, i.e., combing geospatial techniques with Maxent and identifying relevant

factor preferences in establishing priority zones for management actions is necessary. For these

suggestions I recommend these references:

10.https://www.mdpi.com/2071-1050/15/18/13669

11.https://doi.org/10.1016/j.ecoinf.2022.101930

12.https://doi.org/10.1007/s10661-024-12438-z

Reviewer #2: Thank you for thoroughly addressing all the concerns raised in my previous review. After reviewing the revised manuscript, I am satisfied with the changes made. The study is technically sound, with rigorous experiments and appropriate statistical analysis that clearly support the conclusions drawn. The manuscript is also well-written in standard English, with no noticeable errors, making the content easy to understand. I have no further concerns, and I recommend the manuscript for publication.

7. PLOS authors have the option to publish the peer review history of their article (what does this mean?). If published, this will include your full peer review and any attached files.

Reviewer #1: No

Reviewer #2: No

---

## [Author Response · Author response to Decision Letter 1]

26 Sep 2024

Reviewer #1: The authors have partly addressed my concerns. The authors still need to address the comments that I provided for the 'Discussion' section with citing the relevant references provided.

" Discussion requires two subsections in which the limitations of the modeling technique

should be discussed. In addition to this, a small section highlighting the benefits of the applied

modeling techniques, i.e., combing geospatial techniques with Maxent and identifying relevant

factor preferences in establishing priority zones for management actions is necessary. For these

suggestions I recommend these references:

10.https://www.mdpi.com/2071-1050/15/18/13669

11.https://doi.org/10.1016/j.ecoinf.2022.101930

12.https://doi.org/10.1007/s10661-024-12438-z

R/ As per your recommendation, we have revised the discussion section on the limitations and benefits of the MaxEnt modelling technique and have incorporated the three articles you suggested.

4.1. Limitations of the Modeling Technique

The applied modeling approach, which integrates geospatial techniques with MaxEnt, presents several limitations. While MaxEnt is a widely used method for species distribution modeling, particularly for rare species with small datasets, it assumes that the species are at equilibrium with the environment, which might not always be true, particularly in highly dynamic ecosystems like those found in our study area. In the model, the environmental and demographic situations are considered fixed. This limitation can be so important because those factors are very dynamic in nature: human migration, changing land use, and climate variability may change the risk landscape for EVD over time [32, 33-35]. The same model applied to another region would need enormous adjustments to take into account local data and, most probably, different environmental and demographic variables. Moreover, the reliance on available environmental variables might introduce biases due to the limited number of climatic factors included in the model. For instance, [36] highlighted the importance of accounting for land-use changes in addition to climatic variables in species distribution models, suggesting that a more comprehensive dataset could enhance prediction accuracy.

Another limitation is the resolution of the geospatial data. The environmental layers used in MaxEnt were sourced from publicly available datasets, which often offer coarse resolutions. This might limit the precision of habitat suitability predictions in microhabitats where fine-scale environmental variation plays a significant role. Furthermore, as identified in studies such as [37], model outputs are sensitive to the choice of regularization parameters, and incorrect parameter settings can either overfit or underfit the data. Future studies could refine these limitations by incorporating high-resolution environmental layers and more robust parameter selection processes, as noted by [36].

4.2. Benefits of the Applied Modeling Techniques

Despite these limitations, combining geospatial analysis with MaxEnt provided significant insights into the spatial distribution of the target species. MaxEnt's ability to handle small sample sizes is particularly advantageous in regions like ours, where data scarcity is a significant challenge. The integration of geospatial techniques further allowed for a spatially explicit analysis that identified priority zones for conservation actions, a critical step in adaptive management strategies. This aligns with the findings of [37], where MaxEnt successfully predicted the distribution of tick vectors under various environmental drivers, offering essential data for establishing early warning systems.

The key strength of this modeling approach lies in its ability to identify the most relevant environmental factors influencing species distribution. For instance, our model revealed that population density, temperature variability and precipitation were critical factors, corroborating findings from studies like [37, 38-39], where similar variables were shown to significantly influence the spatial distribution. 

In addition, both studies underscore the importance of predictive modeling for future conservation planning. In [40], they used MaxEnt to predict future oak tree distribution ranges under various climate scenarios, providing valuable information for proactive conservation efforts. In a parallel application, [37] employed similar techniques to predict the spatial distribution of Hyalomma ticks, demonstrating the utility of MaxEnt in addressing both plant and animal species distributions across different ecosystems. Together, these studies affirm that MaxEnt, when combined with geospatial techniques, is a powerful tool for both current and future ecological planning.

36. Mirhashemi, H., Ahmadi, K., Heydari, M., Karami, O., & Valkó, O. (2024). Climatic variables are more effective on the spatial distribution of oak forests than land use change across their historical range. Environmental Monitoring and Assessment, 196, 289. https://doi.org/10.1007/s10661-024-12438-z

37. Khwarahm, N. R. (2023). Predicting the spatial distribution of Hyalomma ssp., vector ticks of Crimean-Congo hemorrhagic fever in Iraq. Sustainability, 15(18), 13669. https://doi.org/10.3390/su151813669

38. HamaAmin, B. A., & Khwarahm, N. R. (2023). Predictive mapping of two endemic oak tree species under climate change scenarios in a semiarid region: Range overlap and implications for conservation. Ecological Informatics, 73, 101930. https://doi.org/10.1016/j.ecoinf.2022.101930

---

## [Decision Letter · Decision Letter 2]

29 Sep 2024

SPATIAL MODELING AND ECOLOGICAL SUITABILITY OF EBOLA VIRUS DISEASE IN AFRICA

PONE-D-24-12702R2

Dear Dr. LOMBO,

We’re pleased to inform you that your manuscript has been judged scientifically suitable for publication and will be formally accepted for publication once it meets all outstanding technical requirements.

Kind regards,

Clement Ameh Yaro, Ph.D

Academic Editor

PLOS ONE

Additional Editor Comments (optional):

Reviewers' comments:

Reviewer's Responses to Questions

**Comments to the Author**

1. If the authors have adequately addressed your comments raised in a previous round of review and you feel that this manuscript is now acceptable for publication, you may indicate that here to bypass the “Comments to the Author” section, enter your conflict of interest statement in the “Confidential to Editor” section, and submit your "Accept" recommendation.

Reviewer #1: All comments have been addressed

2. Is the manuscript technically sound, and do the data support the conclusions?

Reviewer #1: Yes

3. Has the statistical analysis been performed appropriately and rigorously? 

Reviewer #1: Yes

4. Have the authors made all data underlying the findings in their manuscript fully available?

Reviewer #1: Yes

5. Is the manuscript presented in an intelligible fashion and written in standard English?

Reviewer #1: Yes

6. Review Comments to the Author

Reviewer #1: The authors have sufficiently addressed my concerns. I believe the manuscript is now ready for publication.

7. PLOS authors have the option to publish the peer review history of their article (what does this mean?). If published, this will include your full peer review and any attached files.

Reviewer #1: No

---

## [Editor Report · Acceptance letter]

10 Oct 2024

PONE-D-24-12702R2 

PLOS ONE

Dear Dr. Baluma Didier, 

I'm pleased to inform you that your manuscript has been deemed suitable for publication in PLOS ONE. Congratulations! Your manuscript is now being handed over to our production team.

Kind regards, 

on behalf of

Dr. Clement Ameh Yaro 

Academic Editor

PLOS ONE